# TRPV4 Stimulation Level Regulates Ca^2+^-Dependent Control of Human Corneal Endothelial Cell Viability and Survival

**DOI:** 10.3390/membranes12030281

**Published:** 2022-02-28

**Authors:** Jennifer Donau, Huan Luo, Iiris Virta, Annett Skupin, Margarita Pushina, Jana Loeffler, Frauke V. Haertel, Anupam Das, Thomas Kurth, Michael Gerlach, Dirk Lindemann, Peter S. Reinach, Stefan Mergler, Monika Valtink

**Affiliations:** 1Institute of Anatomy, Faculty of Medicine, TU Dresden, 01307 Dresden, Germany; jennifer.donau@tu-dresden.de (J.D.); annett.skupin@tu-dresden.de (A.S.); margo171190@gmail.com (M.P.); janaloeffler@web.de (J.L.); 2Institute of Medical Microbiology and Virology, Faculty of Medicine, TU Dresden, 01307 Dresden, Germany; dirk.lindemann@tu-dresden.de; 3Klinik für Augenheilkunde, Charité—Universitätsmedizin Berlin, Corporate Member of Freie Universität Berlin, Humboldt-Universität zu Berlin, and Berlin Institute of Health, 13353 Berlin, Germany; huan.luo@charite.de (H.L.); iiris.virta@hotmail.com (I.V.); 4Institute of Physiology, Faculty of Medicine, University Giessen, 35392 Giessen, Germany; frauke.haertel@physiologie.med.uni-giessen.de; 5Institute of Physiology, Faculty of Medicine, TU Dresden, 01307 Dresden, Germany; anupam.das@tu-dresden.de; 6Center for Molecular and Cellular Bioengineering (CMCB), Technology Platform, TU Dresden, 01307 Dresden, Germany; thomas.kurth@tu-dresden.de; 7Core Facility Cellular Imaging, Faculty of Medicine, TU Dresden, 01307 Dresden, Germany; michael.gerlach2@tu-dresden.de; 8School of Ophthalmology and Optometry, Wenzhou Medical University, Wenzhou 325027, China; preinach25@gmail.com; 9Equality and Diversity Unit, Faculty of Medicine, TU Dresden, 01307 Dresden, Germany

**Keywords:** human corneal endothelial cells, transient receptor potential vanilloid subtype 4, cell survival, cell surface differentiation, transepithelial electrical resistance, intracellular Ca^2+^ signaling

## Abstract

The functional contribution of transient receptor potential vanilloid 4 (TRPV4) expression in maintaining human corneal endothelial cells (HCEC) homeostasis is unclear. Accordingly, we determined the effects of TRPV4 gene and protein overexpression on responses modulating the viability and survival of HCEC. Q-PCR, Western blot, FACS analyses and fluorescence single-cell calcium imaging confirmed TRPV4 gene and protein overexpression in lentivirally transduced 12V4 cells derived from their parent HCEC-12 line. Although TRPV4 overexpression did not alter the baseline transendothelial electrical resistance (TEER), its cellular capacitance (Ccl) was larger than that in its parent. Scanning electron microscopy revealed that only the 12V4 cells developed densely packed villus-like protrusions. Stimulation of TRPV4 activity with GSK1016790A (GSK101, 10 µmol/L) induced larger Ca^2+^ transients in the 12V4 cells than those in the parental HCEC-12. One to ten nmol/L GSK101 decreased 12V4 viability, increased cell death rates and reduced the TEER, whereas 1 µmol/L GSK101 was required to induce similar effects in the HCEC-12. However, the TRPV4 channel blocker RN1734 (1 to 30 µmol/L) failed to alter HCEC-12 and 12V4 morphology, cell viability and metabolic activity. Taken together, TRPV4 overexpression altered both the HCEC morphology and markedly lowered the GSK101 dosages required to stimulate its channel activity.

## 1. Introduction

Corneal transparency and visual acuity are both dependent on the maintenance of the structural and functional integrity of the endothelial monolayer array, which lines the posterior cornea facing the anterior chamber of the eye. These hexagonal cells are attached to the basement Descemet membrane complex. Unlike the corneal endothelium in other vertebrates and in some mammals, the human corneal endothelium (HCE) does not contain mitotically active cells [1]. Under normal conditions, the HCE offsets tissue swelling that can result in losses in transparency. Such declines are attributable to losses in the ability of this cell layer to elicit sufficient outward-directed net fluid transport. Such movement is needed to counterbalance osmotic-driven inward fluid uptake and tissue swelling.

Another feature that contributes to the ability of the endothelial layer to mediate sufficient fluid efflux is its low paracellular junctional resistance [2]. The maintenance of this resistance is dependent on the extracellular presence of calcium, since exposure of the HCE to either a calcium-free solution or a medium containing a calcium channel antagonist disrupts the junctional structural integrity and its permeability and selectivity [3,4]. Such declines lead to rounding up of endothelial cells, followed by their detachment and subsequent rapid corneal swelling. Accordingly, studies focused on characterizing the underlying mechanisms mediating preservation of corneal endothelial functional and structural integrity are relevant to improving therapeutic management of diseases that accompany declines in corneal transparency and losses in visual acuity. 

The maintenance of tissue transparency is also dependent on the ability of the endothelium to express a wide range of ion transporter ATPases, co-transporters and ion channels. Their coordinated activity establishes the ionic gradients across the paracellular and cellular pathways, which underlie osmotically coupled outward fluid egress from the stroma into the anterior chamber. Expression of voltage-operated L-type Ca^2+^ channels (VOCCs) was first suggested by Green et al. [4] and confirmed by Mergler et al. in HCE cells [5]. More recently, expression of different subtypes of the transient receptor potential (TRP) superfamily of ion channels was identified in the HCE. They include the melastatin 8 subtype (TRPM8) [6,7] and TRPs of the vanilloid 1–4 subtypes [8,9]. Several studies have shown that aberrant intracellular calcium transients triggered through VOCCs and TRPs induce apoptosis in different types of cells [10,11]. Notably, TRPV4 is expressed in the HCEC-12 cell line, as well as two clonal daughter cell lines (HCEC-H9C1 and HCEC-B4G12). Furthermore, functional TRPV4 channel expression was identified in both HCEC-12 and in its clonal daughter cell lines by showing that known selective activators induce Ca^2+^ transients and underlying currents that are characteristic of this channel subtype [9]. The functional relevance of TRPV4 expression to maintain tissue transparency has also been shown for the human corneal epithelium [12,13,14,15,16].

TRPV4 is a non-selective calcium-permeant cation channel that acts as an osmosensor of hypoosmolality. Its activation is exemplified by increases in intracellular Ca^2+^ influx that underlie increases in outward ionic rectification [17,18,19]. Since TRPV4 is linked to signaling pathways that control cell proliferation, cell survival, ATP release, and IL-8 production in other cells [20], we determined here whether its upregulation alters viability and survival through modulation of HCE Ca^2+^ homeostasis. In order to shed light on the role of TRPV4 expression levels in mediating responses underlying control of HCE homeostasis, we evaluated the effects of its overexpression on some responses that underlie its normal function in vitro.

## 2. Materials and Methods

### 2.1. Cell Culture

The human corneal endothelial cell population, HCEC-12 [21], and its derivatives transduced with either empty vector (12EV) or TRPV4 (12V4) were cultured in medium F99HCEC (Ham’s F12/Medium 199, 5% FCS, 20 µg/mL ascorbic acid, 10 ng/mL human recombinant FGF-2, 20 µg/mL human recombinant insulin, 2.5 µg/mL amphotericin B and 50 µg/mL gentamycin) at 37 °C in a humidified atmosphere containing 5% CO_2_. Media, supplements and reagents for cell culture were purchased from Life Technologies Invitrogen (Karlsruhe, Germany) or Biochrom AG (Berlin, Germany). Growth medium was changed 3 times per week. Cells were passaged by trypsinization or with Accutase (PAA Laboratories, Pasching, Austria) and routinely seeded at a density of approx. 4000 cells/cm^2^ in T25 culture flasks coated with 10 µg/mL laminin and 10 mg/mL chondroitin sulphate. Cells transduced to overexpress TRPV4 (12V4) were cultured under constant selective conditions with 10 µg/mL puromycin (Applichem, Darmstadt, Germany). 

The human embryonic kidney cell lines HEK293T (ATCC CRL-1573) [22] and HT1080 (ATCC CCL-121) [23] were cultured in DMEM + 10% FCS and 10 µg/mL penicillin/streptomycin at 37 °C in a humidified atmosphere containing 5% CO2. Cells were passaged by trypsinization at a split ratio of 1:10 or 1:20 at confluence.

### 2.2. Cloning and Lentiviral Transduction

To resolve TRPV4 cellular localization and distribution, the transduced TRPV4 gene was labelled with an HA-tag. The sequences of human TRPV4 were isolated from pCR4TOPO TRPV4 (Biocat GmbH, Heidelberg, Germany) and tagged with an N-terminal 27 bp HA-tag sequence produced by oligo annealing (oligonucleotides were ordered from Eurofins, Hamburg, Germany). These tagged sequences were cloned into the lentiviral vector p6NST58, a derivative of p6NST50 [24] that expresses a spleen focus-forming virus U3 promoter-driven tricistronic transcript harboring a 5′ transgenic sequence and a 3′ IRES-controlled puromycin resistance cassette (Puro^R^), followed by a T2A element and a DsRedExpress2 (DsRedEx2) ORF, yielding the expression vector p6NST58-HA-TRPV4. Correct insertion of the constructs was confirmed by control digests and sequencing. The cloning strategy and primers are available upon request. Viral vectors were produced in HEK293T cells with the three-component approach using p6NST58-HA-TRPV4 transfer-vector plasmid, the Gag/Pol encoding plasmid pCD/NL-BH [25] and the simian foamy virus glycoprotein variant encoding plasmid pcoSE03 [26] at a ratio of 7:7:1. Empty vector controls (with p6NST58 as transfer vector) and mock controls (pUC19 transfection supernatants) were produced likewise as controls. Lentiviral vector particles (LVPs) containing 293T supernatants were harvested 48 h after transfection of HEK293T cells with the plasmids, passed through a 0.45 µm syringe filter, quick-frozen in aliquots and stored at −80 °C until further use. 

Target cells were plated at a density of 6.6 × 10^3^ cells/cm^2^ in 12-well plates either 24 h (HT1080) or 48 h (HCEC-12) before transduction. Cells were infected with 1 mL of the respective cell culture medium containing 1:10 to 1:10,000 dilutions of LVP-containing HEK293T cell culture supernatant for 4 h. Cells transduced with empty vector LVPs (EV) and mock transduced cells served as controls. The medium was changed 4 h after infection, and transduction efficiencies were determined by flow cytometry (BD FACSCalibur, Becton Dickinson, Franklin Lakes, NJ, USA) using a fluorescence marker gene transfer assay at 3 days post infection (dpi). Fluorescence readings were taken from 10,000 cells within the defined gate, and data were recorded and processed with BD CellQuest™ and BD FlowJo^TM^ software. Titers were calculated from samples with 0.1% to 80% DsRedEx2 positive cells. To generate stably transduced cell populations, lentivirally transduced HCEC-12 derivatives were plated in 6-well plates and selected with puromycin (10 µg/mL) beginning at 2 dpi. Cells were further cultured with medium changes every other day and eventually passaged and expanded for further analyses.

### 2.3. Quantitative Real-Time PCR

Total RNA was extracted from cell pellets (one well of a 6-well plate each) with the RNeasy Mini Kit (Qiagen, Hilden, Germany), according to the manufacturer’s instructions. These cell pellets contained either non-transduced HCEC-12, empty vector-transduced HCEC-12 (12EV) or TRPV4-transduced HCEC-12 (12V4). Post-extraction RNA samples (250 ng total cellular RNA) were treated for 30 min at 37 °C with 1 U DNase I (Thermo Fisher Scientific, Waltham, MA, USA) in a total volume of 18 µL. DNase I digestion was terminated by adding 2 µL DNase inactivation reagent according to the manufacturer’s protocol, resulting in a total reaction-sample volume of 20 µL. Ten µL of DNase I digested RNA samples were reverse-transcribed or mock-incubated in a total volume of 20 µL using a RevertAid H- reverse transcriptase kit (Thermo Fisher Scientific, Waltham, MA, USA) and an oligo dT30 primer. Subsequently, cDNA aliquots were used in duplicate for qPCR analysis, employing a TRPV4-specific primer–probe set with each 10 pmol/µL Maxima Probe qPCR master mix (2×), including ROX reference dye (Thermo Fisher Scientific, Waltham, MA, USA), in a StepOnePlus real-time PCR system (Applied Biosystems by Thermo Fisher Scientific, Waltham, MA, USA.). Samples were initially denatured for 10 min at 95 °C and subsequently amplified in 40 cycles of 30 s at 95 °C, 30 s at 58 °C and 30 s at 72 °C. All values obtained were referred to a standard curve consisting of 10-fold serial dilutions of a reference plasmid containing the target sequences. All sample values were included that were in the linear range of the standard curve that spanned an interval from 10^1^ to 10^9^ copies per qPCR reaction. Determined qPCR values were expressed as copies/ng total RNA and normalized to the expression level of the endogenous human beta-actin housekeeping control gene. The primer–probe sets used were 5′ GTGGTGCTTCAGGGTGGATG 3′ (forward), 5′ GTCCGGGTTCGAGTTCTTG 3′ (reverse) and 5′ TACCGTGGGCCGCCTCCGCA 3′ (probe) for TRPV4 and 5′ TGGACTTCGAGCAAGAGATG 3′ (forward), 5′ GAAGGAAGGCTGGAAGAGTG 3′ (reverse) and 5′ CGGCTGCTTCCAGCTCCTCC 3′ (probe) for human beta-actin.

### 2.4. Western Blotting

All chemicals were purchased from Carl Roth GmbH + Co. KG (Karlsruhe, Germany) and GE Healthcare GmbH (Solingen, Germany). Cells were cultured in T25 flasks and lysed at confluence in 100 µL protein sample buffer (300 mmol/L Tris/HCl pH 6.8, 10% SDS, 30% glycerol, 0.1% bromophenol blue, 10 mmol/L DTT). The supernatants were sonicated 3 times for 10 s on ice and either stored frozen at −20 °C until further use or directly subjected to Western blotting. Samples were boiled at 95 °C for 10 min, loaded on self-cast 7.5% PAA gels and electrophoresed (running buffer: anode 0.2 mol/L Tris base, pH 8.9; cathode 0.1 mol/L Tris base, 0.1 mol/L Tricine, 0.1% SDS) at 20 mA overnight. Proteins were blotted onto Amersham Hybond™ ECL™ nitrocellulose membranes in a semi-dry blotting cassette (transfer buffer: 48 mmol/L Tris, 39 mmol/L glycine, 0.037% (*w*/*v*) SDS, 20% (*w*/*v*) methanol) at 1.23 mA/cm^2^ for 90 min. Membranes were blocked in 5% milk powder in PBST (0.5% *v*/*v* Tween^®^20 in PBS) at room temperature for 1 h, followed by incubation with mouse anti-HA or rabbit anti-TRPV4 and mouse anti-human γ-tubulin diluted in blocking buffer at 4 °C overnight. Membranes were washed with PBST and incubated with HRP-conjugated secondary antibodies diluted in blocking buffer at RT for 1 h (Table 1). After washing, blots were developed using ECL reagent (Immobilon Western Chemiluminescent HRP Substrate), visualized in a bioimager (LAS3000 imaging system, Fujifilm Europe GmbH, Dusseldorf, Germany) and analyzed with Image Gauge (Version 4.22, Fujifilm Europe GmbH) and Photoshop^®^ (Version 10.0, Adobe^®^) software.

### 2.5. Immunofluorescence Staining

HCEC-12, 12EV and 12V4 cells were seeded at a density of 1 × 10^4^ cells in 400 µL medium per well in 8-well chamber slides and analyzed by immunohistochemical staining after reaching confluence. Unstained cells, as well as those incubated with secondary antibody only or isotype immunoglobulin, served as controls. Cells were fixed with 4% formaldehyde for 20 min and permeabilized with 0.5% saponin in PBS for 30 min for visualization of DsRedEx2 fluorescence or were fixed with ice-cold methanol for 20 min for immunofluorescent double staining. Fixed cells were rinsed 3 times in PBS for 10 min and blocked in 1% BSA in PBS at RT for 1 h. Samples were stained with primary antibodies (Table 1) diluted in PBS + 1% BSA and incubated at RT for 1 h. Samples were washed again with PBS and incubated with the secondary antibodies (Table 1) at 4 °C for 1 h. Nuclei were stained with Hoechst 33342 (1:1000; Invitrogen), and slides were mounted with glycerol/DABCO. Staining was visualized and photo-documented under an Olympus IX70 fluorescence microscope equipped with an UPlanSApo 40× objective and an F-View CCD camera run by analysis^®^ software (Software Imaging System GmbH, Muenster, Germany) for visualization of DsRedEx2 or under a Zeiss LSM 880 microscope equipped with an LD LCI Plan-Apochromat 40×/1.2 Imm Korr DIC M27 objective, a Quasar detector (GaAsP PMT 32 channel spectral detector) and an Airyscan detector in “Superresolution” mode at zoom factor 7 and master gain of 800 run by ZEN software (Carl Zeiss Microscopy GmbH, Jena, Germany) for immunofluorescent double staining and z-stacks (glycerol immersion, voxel size 73 nm × 73 nm × 277 nm). 

For flow cytometric analyses, HCEC-12, 12EV and 12V4 cells were seeded at a density of 1.25 × 10^5^ cells per T25 flask and grown to sub-confluence. Then, the cells were harvested, washed and centrifuged at 300× *g* for 5 min and fixed in 1 mL ice-cold methanol. Fixed cells were washed in PBS and blocked in 1% BSA/PBS at RT for 1 h. After centrifugation at 350× *g* for 5 min, the cells were stained with 100 µL of primary antibody in 1% BSA/PBS at RT for 1 h or overnight at 4 °C, washed and incubated with 100 µL of secondary antibody in 1% BSA/PBS at 4 °C for 1 h. Cells were analyzed by flow cytometry on a BD FACSCalibur as described above.

### 2.6. Fluorescence Calcium Imaging

HCEC-12 and 12V4 cells were seeded on coverslips at a density of 4000 cells/mm^2^ and grown to confluence. Cells were loaded with 1 µmol/L fura-2-AM in culture medium at 5% CO_2_ and 37 °C for 40 min, followed by rinsing in a Ringer-like (control) solution containing (mmol/L): 150 NaCl, 6 CsCl, 1 MgCl_2_, 10 glucose, 10 HEPES and 1.5 CaCl_2_ at pH 7.4 and 317 mOsmol/L [27]. Changes in intracellular free Ca^2+^ ([Ca^2+^]_i_) were determined based on measurements of fura-2 fluorescence signals at room temperature (23 °C) with a digital camera and software (Olympus Europa Holding GmbH, Hamburg, Germany). Cells on 15 mm diameter coverslips were bathed in the Ringer-like solution for measurement. Single cells were chosen as regions of interest and marked for further analyses. Cells were alternately exposed to 340 nm and 380 nm UV light within a 5 s loop using an LED light source (LED-Hub by Omikron, Rodgau-Dudenhofen, Germany). The fluorescence ratios were calculated of images simultaneously recorded with a digital camera (Olympus XM-10) and f_340nm_/f_380nm._ Their values are proportional to [Ca^2+^]_i_ and were calculated by cellSens software (Olympus Europa Holding GmbH, Hamburg, Germany). Fluorescence ratios were normalized (control set to 0.1) and averaged (with error bars), and results are shown as mean traces of the f_340nm_/f_380nm_ ratio ± SEM (error bars in both directions), with n-values indicating the number of experiments per data point. Delays of [Ca^2+^]_i_ increases in some experiments were due to drugs being pipetted into a stationary bath rather than a flow-through system. When using TRP4 channel modulators, cells were pre-incubated with the channel modulators for approximately 30 min before performing the measurement. Drugs were dissolved in dimethyl sulfoxide (DMSO) to obtain a stock solution and diluted to provide a working concentration that did not exceed 0.1% DMSO. This DMSO concentration was nontoxic, based on stable f_340nm_/f_380nm_ levels (data not shown).

### 2.7. Cell Viability Assays

Metabolic activity was evaluated by resazurin conversion measurements to determine whether transduction and/or transgene expression had any cytotoxic effects. This assay evaluates the capability of the cells´ redox systems to convert blue, non-fluorescent resazurin to pink, fluorescent resorufin during cell metabolism. The resulting fluorescence intensity is a direct measure of the overall metabolic activity in the culture. Its utilization provides a meaningful assessment of drug cytotoxicity. HCEC-12, 12EV and 12V4 cells were seeded at a density of 3 × 10^3^ cells in 100 µL medium per well in 96-well plates and allowed to attach and spread for 72 h. After 3 d, TRPV4 channels were stimulated in all three populations with the TRPV4 activator GSK101 (1 pmol/L–10 µmol/L) or with the TRPV4 inhibitor RN1734 (1–30 µmol/L) for 24 h. As solvent controls, other HCEC-12, 12EV and 12V4 cells were treated with respective volumes of solvent (DMSO). After 22 h of incubation with TRPV4 channel modulators or solvent, 20 µL resazurin was added per well containing 100 µL medium, and the cells were incubated for another 2 h. Conversion of blue resazurin to pink-colored fluorescent resorufin was measured fluorometrically in a Tecan infinite M200 plate reader (Tecan, Crailsheim, Germany) using a 560/590 nm excitation/emission filter pair. Medium with resazurin in cell-free wells served as a blank. Data were recorded and processed with i-control software (Version 1.6.19.2) by Tecan and GraphPad Prism (Version 8.2.1) software. The mean blank value was subtracted from each sample value, and mean values were expressed as mean ± SEM.

HCEC-12, 12EV and 12V4 cells were seeded at a density of 1 × 10^5^ cells in 1 mL medium per well in 12-well plates. TRPV4 channel activity was modulated as described above. After 24 h, cells were collected by trypsinization, combined with floating cells from the respective wells, centrifuged at 300× *g* for 5 min with moderate braking and resuspended in 100 µL PBS + 1% FCS prewarmed to 37 °C. Vital staining was performed with 2 nmol/L SYTOX^TM^ Green nucleic acid stain (Thermo Fisher Scientific, Waltham, MA, USA), which is impermeant to living cells but penetrates dead cells with compromised membranes and, by binding to nucleic acids, emits a green fluorescence peaking at 523 nm. After 20 min of incubation in the dark at room temperature, fluorescence readings of samples were immediately acquired using a FACSCalibur flow cytometer with a 530 ± 30 nm filter. At least 10,000 events were collected for each sample. Multiparameter data analysis was performed with FlowJo^TM^, and the percentages of dead (SYTOX^TM^ Green positive) cells were calculated as mean ± SEM.

### 2.8. Transepithelial Electrical Resistance (TEER) and Cellular Capacitance (Ccl) Measurement

A cellZscope impedance spectroscope (nanoAnalytics, Münster, Germany) was used to analyze TEER and Ccl of confluent monolayers of HCEC-12, 12EV and 12V4 cells cultured on Transwell^®^ polycarbonate membrane well inserts (6.5 mm diameter, 0.4 µm pore size; Corning Incorporated, Corning NY, USA) [28]. After obtaining baseline recordings with a low alternating current of 1 Hz–100 kHz for 24 h, the medium was replenished with or without different concentrations of GSK101, as described above, and TEER and Ccl were recorded for another 48 h. Measurements were carried out at 37 °C in a humidified atmosphere containing 5% CO_2_. The background signal was recorded with cell-free membrane well inserts to determine TEER and Ccl using nanoAnalytics software.

### 2.9. Scanning Electron Microscopy

Non-transduced HCEC-12 and its transduced derivatives 12EV and 12V4 cells were cultured on 12 mm diameter Thermanox^TM^ coverslips (Thermo Fisher Scientific) to confluence. Cells were then fixed in 1% glutaraldehyde in 100 mmol/L phosphate buffer and washed first in buffer (2×), then in water (4×). Samples were postfixed in 1% osmium tetroxide in water, washed several times in water, dehydrated in an ascending ethanol concentration series (30, 50, 70, 90, 96% ethanol, 3 × 100% ethanol on molecular sieve) and critical-point-dried using a Leica CPD300 drier (Leica Microsystems, Wetzlar, Germany). Samples were mounted on 12 mm aluminum stubs, sputtered with gold (60 mA, 60 s) and analyzed with a Jeol JSM 7500F cold field-emission scanning electron microscope running at 5 kV and with a working distance of 8 mm (Jeol Germany GmbH, Freising, Germany).

### 2.10. Statistical Analysis

Graph Pad Prism (La Jolla, CA, USA) was used to calculate FACS, cell viability assay, TEER and Ccl results, and the recorded values are expressed as mean ± SEM with error bars in both directions. Data were grouped according to treatment (TRP channel modulators). Statistical analysis was performed by one-way ANOVA, followed by Tukey multiple comparisons test. For cell volume determinations, statistical analysis by Brown–Forsythe and Welch ANOVA tests with Games–Howells multiple comparisons was performed. For fluorescence data analyses, statistical significance was determined with parametric Student’s *t*-test for paired and unpaired data if the data passed the normality test. Otherwise, the non-parametric Wilcoxon test was used. Significance was accepted at *p* ≤ 0.05. Plots were generated with SigmaPlot (Systat Software, San Jose, CA, USA).

## 3. Results

### 3.1. TRPV4 Channel Overexpression in HCEC-12

HCEC-12 and the control cell line HT1080 were transduced with LVPs encoding for human TRPV4 (Figure 1a), resulting in 51% and 60% transduced cells, respectively, as indicated by DsRedEx2 expression at 3 dpi (Figure 1b,c). After infection of target cells, flow cytometric titration analysis of produced LVPs containing supernatants for stable DsRedEx2 expression revealed an approximately five to eightfold higher infection titer for EV LVPs compared with V4 LVPs. The percentage of DsRedEx2 positive cells was augmented by selection with puromycin, which stabilized at 95% one week after selection. To promote invariant transgene expression, cells were kept under constant puromycin selection pressure.

Quantitative real-time PCR analysis with exon-spanning primers for the detection of all TRPV4 transcript variants showed that TRPV4 was expressed at a level that was 1000-fold higher (*p* = 0.001) in 12V4 cells compared to the parent HCEC-12 counterpart (Figure 2a). FACS analysis confirmed higher TRPV4 protein expression levels after immunostaining either indirectly against the HA-tag or directly against TRPV4 in the 12V4 cultures. The results show that the percentage of TRPV4-overexpressing cells increased to >90% in puromycin-selected 12V4 cultures. Mean fluorescence intensity (MFI) results after immunostaining of the three HCEC cell lines showed a three to eightfold higher TRPV4 protein production in TRPV4-transduced cells (12V4) compared to transduction controls (12, 12EV) (Figure 2b,c). FACS data on HA-TRPV4 protein production were further confirmed by Western blotting with antibodies against the HA-tag, revealing vast amounts of HA-TRPV4 in 12V4 cells (Figure 2d). Western Blot analysis with antibodies against TRPV4 (Figure 2e) indicated two different protein bands that were also found in native HCEC-12 and corresponded to those seen in the HA-TRPV4 Western Blot. One band appeared at approx. 100 kDa, corresponding to the 98 kDa canonical form, and the other band appeared at approx. 115 kDa, representing an unprocessed form, confirming that the two bands in the HA-TRPV4 Western blot represent TRPV4 expression.

TRPV4-overexpressing cells had a normal, corneal endothelial-like phenotype with a polygonal morphology when viewed under phase contrast. As in non-transduced cells, Na^+^,K^+^-ATPase was localized laterally, and cells had an interrupted band of ZO-1-positive tight junctions at their lateral borders, as determined by immunocytochemistry. The HA-tagged TRPV4 protein localized predominantly to the cell membrane, as there were no signals overlapping with detectable, cytoplasmic DsRedEx2 fluorescence. Extensive colocalization of HA-TRPV4 immunopositivity with ZO-1 or Na^+^,K^+^-ATPase immunopositivity was not detected, but colocalization cannot be fully excluded (Figure 3, Appendix A).

### 3.2. Functional Overexpression of TRPV4

GSK101 is a highly selective and potent TRPV4 channel agonist that has been used to confirm TRPV4 overexpression in 12V4 cells [29,30]. Since TRPV4 has already been described in HCEC-12 cells [9], the TRPV4-induced Ca^2+^ influx was compared between normal HCEC-12 and the 12V4 cell population. The TRPV4-induced Ca^2+^ influx in the 12V4 cells was at clearly higher levels (Figure 4a). The f_340nm_/f_380nm_ fluorescence ratio increased from 0.199 ± 0.001 (control; t = 100 s) to 1.86 ± 0.107 (t = 350 s; *n* = 26; *p* < 0.005) in 12V4, whereas its value at the same time point for the non-transduced HCEC-12 rose only to 0.2 ± 0.0002 and 0.246 ± 0.007 (10 µmol/L GSK101; *n* = 26; *p* < 0.005) (Figure 4b). This large difference in responses to GSK101 confirmed functional TRPV4 overexpression in 12V4 cells.

### 3.3. TRPV4 Overexpression Diminishes Cell Survival

Phase-contrast microscopic evaluation of the transduced 12V4 cell line cells showed that they formed a regularly arranged monolayer with cell-layer integrity similar to non-transduced counterparts (Figure 5a). Although inhibiting TRPV4 with RN1734 had no effect on cell morphology and viability, GSK101 at a concentration as low as 1 nmol/L induced appreciable concentration-dependent increases in the number of dead 12V4 cells. In contrast, HCEC-12 could tolerate a 1000-fold higher concentration of 1 µmol/L GSK101 without showing any obvious signs of declining cell viability. The cell-layer integrity at 1 µmol/L GSK101 was disrupted, and almost all cells were detached from the surface, with large amounts of cellular debris floating in the 12V4 culture medium. This increase in sensitivity to GSK101 in the 12V4 cells suggests that the TRPV4 expression level is a variable determining cell viability.

The resazurin conversion assay results provided a more detailed analysis of the effect of TRPV4 overexpression on metabolic activity. Lentiviral transduction with a transgene-free transfer vector did not affect basal metabolic rates of 12EV cells compared to the parent HCEC-12. However, overexpression of TRPV4 reduced the basal metabolic rate in 12V4 cells to levels that were 1.5-fold and 1.3-fold lower than those in HCEC-12 and 12EV cells, respectively (*p* = 0.001, Figure 5b). TRPV4 channel activation by GSK101 in the picomolar range (i.e., 0.01 to 0.1 nmol/L) increased metabolic activity in all three cell populations. On the other hand, treatment of the control 12EV cells with GSK101 concentrations ranging from 1 nmol/L to 10 µmol/L did not significantly affect their metabolic rates. In contrast, in 12V4 cells, GSK101 concentrations higher than 0.1 nmol/L were cytotoxic. One nmol/L GSK101 decreased the 12V4 metabolic activity 12.1-fold, and concentrations ≥ 10 nmol/L ceased their metabolic activity compared to the untreated counterpart. Instead, 10 µmol/L GSK101 was required to reduce the metabolic activity of HCEC-12 cells by 3.9-fold compared to their untreated counterpart. In contrast, treatment with the TRPV4 antagonist RN1734 (1–30 µmol/L) did not significantly affect the metabolic activity of any of the three cell lines (Figure 5c).

Vital staining with SytoxTM Green revealed a similar basal level of necrotic cells in all three HCEC populations, with 3.8% necrotic cells in non-transduced HCEC-12, 4.7% in 12EVand 3.8% in the TRPV4-overexpressing 12V4 cells (Figure 5d). However, GSK101 treatment had markedly different effects on necrosis in the three cell lines. Activation of TRPV4 for over 24 h with 1 to 100 nmol/L dose-dependently increased the number of necrotic cells in 12V4 (1 nmol/L: 21.3% ± 1.4%; 10 nmol/L: 70.8% ± 2.8%; 100 nmol/L: 79.4% ± 1.3%), and these declines were invariant following exposure to concentrations higher than 100 nmol/L GSK101. In contrast, the control HCEC-12 and 12EV cell populations tolerated concentrations that were 1000-fold higher than those required to induce the highest level of cytotoxicity in the transduced 12V4 cells. In the non-transduced HCEC-12, TRPV4 10 µmol/L GSK101 was required to increase the percentage of dead cells by only 8.3-fold, which caused the percentage of necrotic cells to increase to 31.7% ± 2.9%. Again, RN1734 did not have a significant inhibitory effect on cell survival of either HCEC-12, 12EV or 12V4 cells (Figure 5e).

### 3.4. Cellular Capacitance and Transepithelial Electrical Resistance

In the absence of GSK101, the 12V4 Ccl was consistently significantly higher than in the HCEC-12 and 12EV cell populations (Figure 6a,b). This difference is in agreement with the enlarged 12V4 and 12EV cell surface areas, which were attributable to either swelling or alterations in cell surface structure. In contrast, the TEER values of the three cell populations were similar to one another for a period lasting 48 h (Figure 6c,d). Exposure to GSK101 for 24 h induced concentration-dependent declines in TEER that were markedly different across the three populations. In 12V4 cells, TEER was reduced at GSK101 concentrations above 1 nmol/L and eliminated by exposure to ≥100 nmol/L GSK101 (Figure 6e). In contrast, in 12EV and HCEC-12 cells, TEER values were much more resistant to increases in GSK101 concentration and less inhibited by GSK101 stimulation. In these two cell populations, the TEER values were relatively invariant despite being exposed to GSK101 at concentrations of up to 1 µmol/L for 24 h.

### 3.5. TRPV4 Overexpression Alters Cell Surface Ultrastructure but Not Cell Volume

Scanning electron microscopy of changes in the apical cell surface integrity showed that there were marked differences between the three cell populations (Figure 7). Whereas the HCEC-12 (Figure 7a–d) and 12EV cells (Figure 7e–h) had rather smooth surfaces with only a few villus-like membrane protrusions, the 12V4 cells (Figure 7i–l) had numerous villus-like membrane protrusions, which were not only more abundant but also much longer than in the HCEC-12 and 12EV cells. This difference in apical surface morphology is also apparent in less confluent or single cells (Figure 7d,h,l). This similarity indicates that the effect is not dependent on epithelial junctional integrity and continuity. Notably, these villus-like protrusion structures were also evident in fluorescence micrographs of DsRedEx2-labelled live cells in z-stacks (Appendix A). There was no distinct difference between 12EV (*n* = 6) and 12V4 (*n* = 5) cells regarding volume or sphericity. However, cells of the non-transduced parent HCEC-12 had a considerably larger volume (*n* = 4) and lower sphericity (*n* = 4), as the heights were comparable between the three cell lines (Appendix A). These differences in cell-line ultrastructural phenotypes suggest that TRPV4 overexpression alters the structural integrity of HCE cells.

## 4. Discussion

We showed that TRPV4 overexpression compromises HCE homeostasis by altering its structural and functional integrity. TRPV4 was chosen as a target for this study because this osmosensitive, thermosensitive and mechanosensitive channel is expressed in the HCE and other ocular tissues [9,12,31,32,33]. This choice is relevant to other tissues expressing TRPV4 because the HCE encounters similar types of stresses that stimulate channel activity in these other tissues [34,35]. Another reason prompting our study is that there are reports showing that variations in TRP channel expression levels affect tissue homeostasis [36,37]. Our assessment involved determining whether TRPV4 overexpression compromises corneal endothelial viability and survival of the well-recognized HCEC-12 line. We showed for the first time that upregulation of TRPV4 expression compromises retention of structural and functional attributes that assure maintenance of corneal homeostasis. Specifically, we showed that the retention of normal HCE function and structure is also dependent on the level of TRPV4 expression because overexpression induces alteration of structural integrity and heightens drug sensitivity to losses in cell viability.

TRPV4 overexpression was achieved by lentiviral transduction in HCEC-12 to create the 12V4 cell population. As in the non-transduced HCEC-12, TRPV4 predominantly localized to the 12V4 cell membranes. Although TRPV4 interacts with the Na^+^,K^+^-ATPase molecular complex in astrocytes [38], we did not observe any colocalization of TRPV4 with either Na^+^,K^+^-ATPase or ZO-1, which are both essential for establishing the leaky barrier and fluid-pumping function in the corneal endothelium. Western blot analysis against the HA-tag of the TRPV4 transgene and against the TRPV4 protein displayed TRPV4 protein band patterns in 12V4 cells that were similar to those in the non-transduced HCEC-12, indicating the presence of the same TRPV4 isoforms in the TRPV4-overexpressing 12V4 cells. Discrepancies in gene and protein expression patterns between qPCR mRNA results and those obtained from flow cytometry and Western blotting may be attributable to a complex time-dependent regulation of TRPV4 expression levels in HCEC.

TRPV4 overexpression in 12V4 cells was confirmed by showing that GSK101 induced much larger rises in calcium influx than in the parent HCEC-12 cells without altering their time-dependent response patterns. These larger transients fell to a baseline level that was higher than that in non-transduced cells, indicating compromise of restorative Ca^2+^ efflux mechanisms. These larger rises in Ca^2+^ influx in the TRPV4-overexpressing 12V4 cells provide an index for assessing the impact of a change in intracellular Ca^2+^ regulation on HCE survival and function.

TRPV4 activation may trigger either apoptotic or necrotic cell death because of perturbation of intracellular Ca^2+^ compartmentalization or intracellular Ca^2+^ overload [39].Variations in TRPV4 expression and activity may determine whether this ionic channel supports cell survival or instead induces cell demise. Therefore, we compared the effects of increasing TRPV4 channel activity with GSK101 or inhibiting it with RN1734 on cell viability and metabolic activity in the 12V4 cells. TRPV4 overexpression increased cell sensitivity towards stimulation but did not compromise cell viability per se. TRPV4 activation with GSK101 impaired HCEC-12 viability and eventually induced cell death. These effects are consistent with the finding that the baseline levels reached following a GSK101-induced transient were considerably higher in 12V4 cells than in the non-transduced HCEC-12. The results are also in agreement with the finding of another study that excessive Ca^2+^ influx mediated by TRPV4 activation predisposed retinal ganglion cells to activate their Ca^2+^-dependent proapoptotic signaling pathways [40]. Taken together with the data from cell survival analyses, it is tenable that regulation of intracellular Ca^2+^-dependent cell survival in 12V4 cells is more easily activated than in the HCEC-12. In addition, TRPV4 overexpression in 12V4 cells markedly increased their sensitivity to GSK101 activation at approximately a 1000-fold lower GSK101 dose than that required to induce a comparable response in HCEC-12 cells.

In other tissues, TRPV4 can mediate apoptosis through its interaction with partners, such as MLC1 [38], most of which are involved in adhesion, proliferation or endocytosis but not expressed in HCE cells. Despite their absence, TRPV4 modulates apoptosis through other mechanisms that may involve Na^+^/K^+^-ATPase beta-1 transporting subunit (ATP1B1), which is expressed in HCE. This active ion-transport mediator is alleged as one of the target loci in Fuchs endothelial dystrophy (FECD), a disease characterized by compromised HCE cells and consequential corneal edema [41]. Whether the observed degeneration of HCE cells in FECD is merely due to an impaired function of ATP1B1 or also to a disturbed interplay with TRPV4 remains to be elucidated. Therefore, TRPV4 osmosensing capability described in numerous other cell types cannot exclusively account for how TRPV4 modulates apoptosis in the corneal endothelium.

A recent study by Jie et al. showed that intraventricular injection of GSK101 activated TRPV4 in a mouse model of ischemic brain infarction and induced apoptosis in the hippocampus. TRPV4 activation downregulated PI3K/Akt and upregulated p38 MAPK signaling pathways and decreased the Bcl-2/Bax ratio. In contrast, blocking TRPV4 activation attenuated brain infarction after occlusion and reperfusion of the middle cerebral artery [42]. These results provide in vivo evidence that the activation of TRPV4 can result in apoptosis in neural tissue. This finding is in agreement with the fact that GSK101-induced in vitro TRPV4 activation markedly increased 12V4 sensitivity to cell death and lowered the required GSK101 dosage to elicit a large decline in TEER.

A microscopic analysis of cell shape and size revealed that the non-transduced HCEC-12 population differed from the transduced 12EV and 12V4 populations in volume and sphericity. This difference suggests that HCEC-12 cells might be composed of a more conductive cytoplasm and more distensible cell membranes. Because quantification of cell volume is difficult in the confluent state without clear boundary markers, our employed quantification method can potentially be biased towards reliably segmented rather than fully connected or confluent cells. This potential error applies to assessing volume changes in all three cell lines even though analysis was performed by the same blind operator to reduce experimental bias. Furthermore, the three investigated cell populations had comparable TEER values, indicating factor uniformity in resistance modulation. Despite comparable TEER values, capacitance (Ccl) of TRPV4-overexpressing cells was significantly higher than that of the non-transduced or empty vector-transduced cell populations. This difference suggests that these cells underwent cell surface enlargement due to changes in cell surface structure, since the cells had a comparable volume to that of 12EV cells.

More extensive characterization of this cell surface enlargement with electron and confocal fluorescence microscopy revealed massive elaborations of villous-like cell protrusions in the transduced 12V4 cells. As described by Yang et al., TRPV4 is involved in invasive cancer cell motility and, in this context, may promote the formation of membrane protrusions, whereas Mrkonjic et al. showed that a thermo-/osmosensing defective TRPV4 mutant, but not wild-type TRPV4, enhances protrusion formation and cell migration [43,44]. Another study described that TRPV4 overexpression leads to accession with the cilia of a variety of human or other mammalian and vertebrate ciliated cells, such as cholangiocytes, oviduct cells or the primary cilium of polarized MDCK cells. This result suggests the involvement of ciliary mechanosensation as another mediator of TRPV4 activation [18].

TRPV4 mechanosensitivity seems to be dependent on its association with cilia in a variety of cells and tissues, e.g., TRPV4 contributes to ciliary beating frequency and mechanotransduction in oviduct cells [45], mesenchymal stem cells [46] and airway epithelia [47]. A primary cilium can also be found in vertebrate and (developing) mammalian corneal endothelial cells and is presumed to play a role in corneal endothelial wound repair after mechanical injury. However, the primary cilium of corneal endothelial cells was also shown to either protrude or retract as a reaction towards environmental changes or stress. On the other hand, it may be involved in sensing the ionic composition or hydrostatic pressure of aqueous humour [48,49,50,51]. In glaucoma, it was suggested that impaired mechanosensing of TRPV4 in trabecular meshwork cells with compromised primary cilia contributes to increased intraocular pressure [12,52]. In this context, it is possible to speculate that a mechanosensing-induced TRPV4 activation may contribute to corneal endothelial cell death in hypertensive glaucoma patients. However, a recent study by Lapajne et al. described only osmo- and thermosensing functional roles for TRPV4 in murine corneal epithelial cells [13], and this pattern may also apply to TRPV4 in the corneal endothelium. Overall, future studies in this area will contribute to a better understanding of HCE biology, and their outcome will hopefully identify novel strategies for prolonging and improving corneal endothelial function in the diseased or medicated eye.

## 5. Conclusions

These results show that the level of TRPV4 expression affects the ability of HCE to survive and function despite being exposed to stresses that challenge sustainment of corneal homeostasis. As TRPV4 overexpression compromises responses underlying survival and HCE functional competence, it may be prudent to monitor TRPV4 expression levels as an indicator of therapeutic efficacy in alleviating losses in corneal transparency resulting from HCE dysfunction.

## Figures and Tables

**Figure 1 membranes-12-00281-f001:**
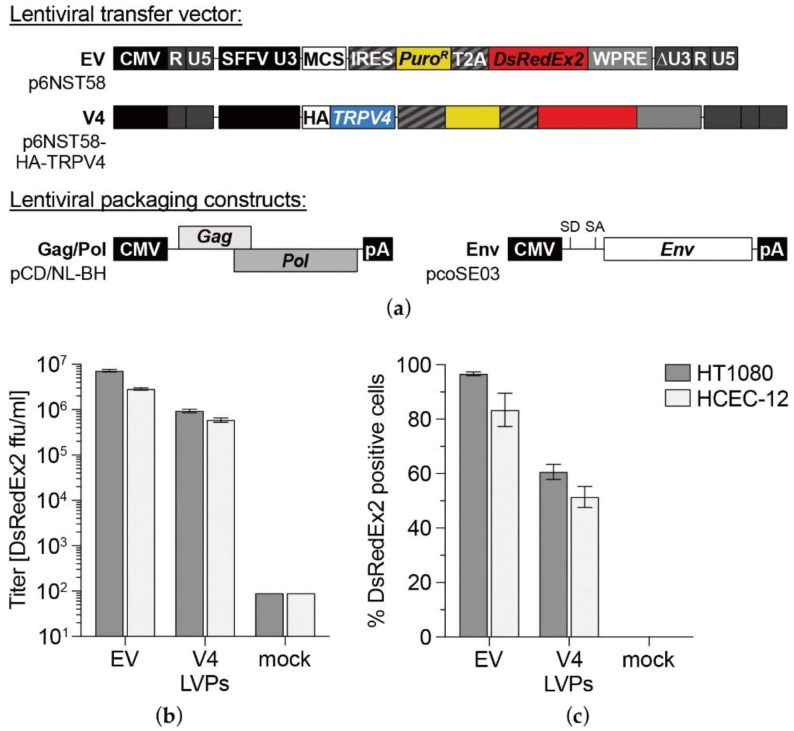
Transduction efficiency of LVPs produced in target cells. Schematic outline (**a**) of the structural organization of lentiviral transfer vectors based on p6NST58 and packaging components (Gag/Pol and Env) for LVP production. Infectious titers [DsRedEx2 ffu/mL] of LVPs (**b**) and percentage of DsRedEx2 positive cells (**c**) as determined by FACS analysis of transduction efficiency in HCEC-12 and the transduction control cell line HT1080. Both cell lines were infected with different dilutions of viral particles containing HEK293T cell culture supernatants. At 3 days post infection (dpi), cells were analyzed by marker gene transfer assay, and titers were calculated. Mean ± SEM from *n* = 3. EV: empty vector LVPs, V4: TRPV4 sequence-carrying LVPs, mock: LVP-free supernatant of pUC19-transfected HEK293T.

**Figure 2 membranes-12-00281-f002:**
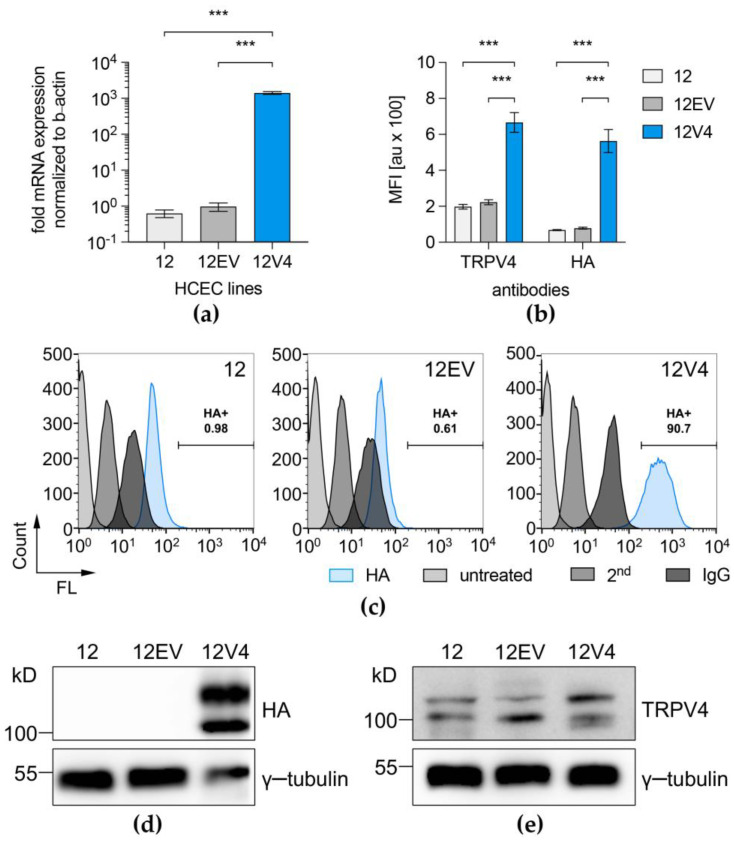
Gene expression and protein expression levels of TRPV4 in the HCEC-12 cell line (12) and its derivatives, 12EV and 12V4. QPCR (**a**) revealed successful TRPV4 mRNA overexpression after transduction (12V4, blue column) compared to controls (light and dark grey columns). Data are presented as the means ± SEM of *n* = 12 (12V4) or *n* = 8 (12EV, 12). TRPV4/HA-TRPV4 protein expression was confirmed by FACS analysis of mean fluorescence intensity (MFI) (**b**) and rates of positively labelled cells in puromycin-selected 12V4 cultures (**c**) after immunostaining of the three HCEC-12 cell lines for indirect (HA) and direct TRPV4 protein determination. Data are presented as the means ± SEM of *n* = 4 (anti-TRPV4 staining) or *n* = 5 (anti-HA staining). The appropriate staining controls, which were used to ensure correct interpretation of the obtained data, namely untreated cells (untreated), secondary antibody control (2nd) and IgG isotype control (IgG), are also shown. Western blot analysis showed abundant HA-TRPV4 (**d**) and native TRPV4 (**e**) protein expression levels in TRPV4-transduced cells. Gamma-tubulin exhibited loading control equivalence. *** *p* ≤ 0.0001.

**Figure 3 membranes-12-00281-f003:**
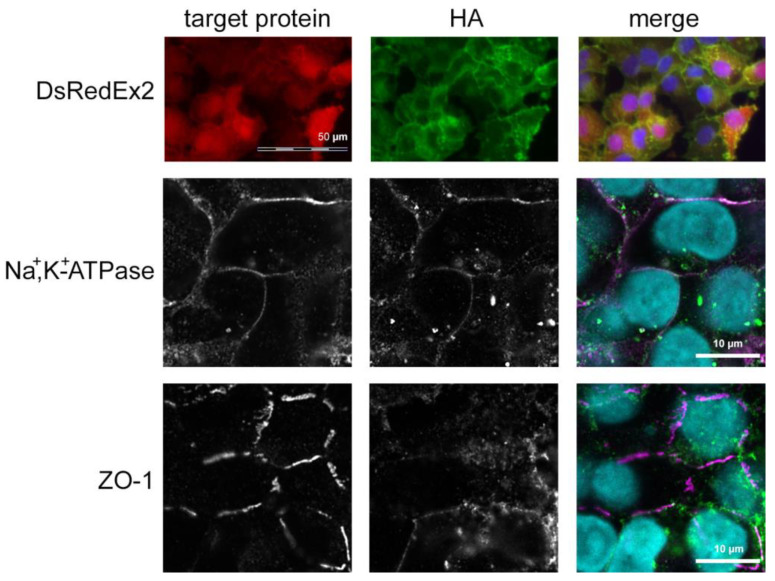
Immunocytochemical staining against the HA-tag and Na^+^,K^+^-ATPase or ZO-1. Immunofluorescence micrographs taken from two exemplary cultures of 12V4 cells. HA: immunocytochemical staining against the HA tag (green fluorescence); target protein: red fluorescence of the protein indicated on the left for each row (DsRedEx2, Na^+^,K^+^-ATPase or ZO-1); merge: combined fluorescence of the HA-tag (green), the target protein (red) and nuclei staining with Hoechst 33342 (blue). Scale bars: 50 µm for upper-row images, 10 µm for middle and lower-row images.

**Figure 4 membranes-12-00281-f004:**
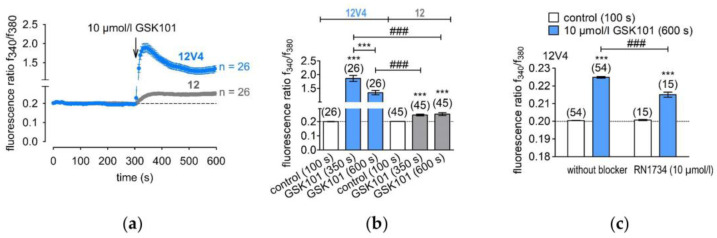
TRPV4 mediated calcium influx in non-transduced HCEC-12 (12) and TRPV4-overexpressing (12V4) cells. (**a**) Time course of the fluorescence ratio (f_340nm_/f_380nm_) of 12V4 (±SEM; *n* = 26; blue trace) and HCEC-12 cells (±SEM; *n* = 26; grey trace). GSK101 (10 µmol/L) was applied after 4 min (arrow). (**b**) Summary of the experiments with GSK101 in 12V4 cells and HCEC-12; *** denotes a significant increase in [Ca^2+^]_i_ in the presence of GSK101 (t = 350 s and 600 s) (±SEM; *n* = 26; *p* < 0.005; paired tested) compared to control (t = 100 s); ### indicates a statistically significant difference in fluorescence ratios of the GSK101 effect in 12V4 cells and HCEC-12 (±SEM; *n* = 26; *p* < 0.005; non-paired tested). (**c**) Summary of the experiments with GSK101 and RN1734 in 12V4 cells; *** denotes a significant increase in [Ca^2+^]_i_ in the presence of GSK101 (t = 600 s) (±SEM; *n* = 54; *p* < 0.005; paired tested) compared to control (t = 100 s); ### indicates a statistically significant difference in fluorescence ratios of the GSK101 effect in 12V4 cells with and without RN1734 (±SEM; *n* = 15–54; *p* < 0.005; non-paired tested).

**Figure 5 membranes-12-00281-f005:**
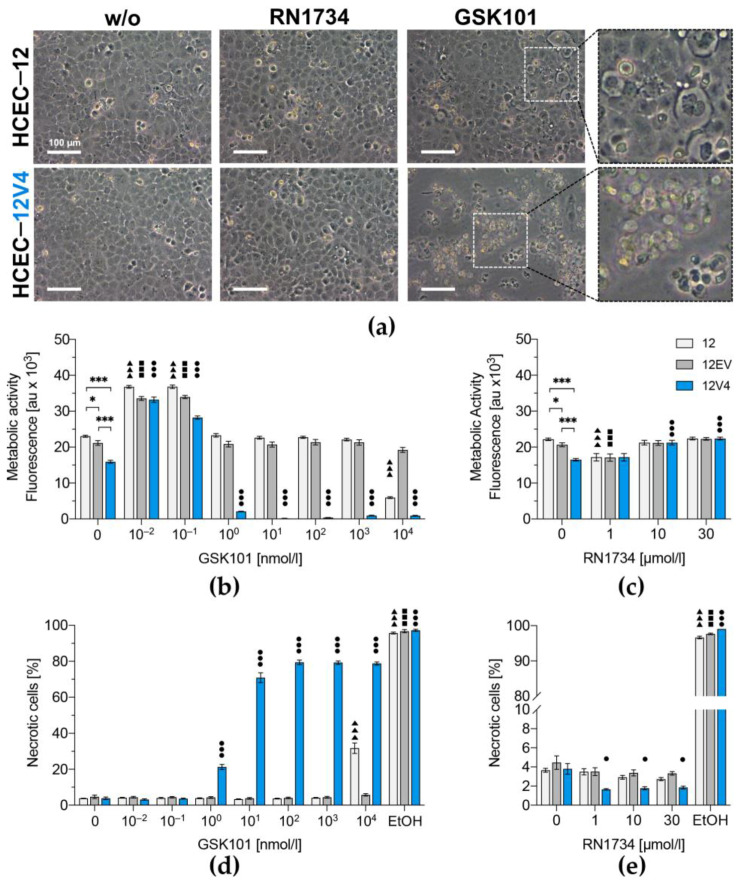
Monitoring of TRPV4-mediated cell death in HCEC-12 (12) and its derivatives, 12EV and 12V4. Morphology of HCEC-12- and HA-TRPV4-transduced HCEC-12 (**a**) either untreated (**left**) or treated with the TRPV4 antagonist RN1734 (**middle**) or the TRPV4 agonist GSK101 (**right**). Differential interference contrast (DIC) micrographs; scale bar 100 µm. Metabolic activity of GSK101- (**b**) or RN1734- (**c**) treated cells were analyzed by resazurin conversion assay. Cells were treated with GSK101 (**d**) or RN1734 (**e**), and membrane integrity was quantified by flow cytometry after vital staining with SYTOX Green to assess cell death. Triangles (▲) represent statistically significant differences compared to the untreated control sample, HCEC-12 (12); squares (◼) represent statistically significant differences compared to untreated HCEC-12EV (12EV); circles (●) represent statistically significant differences compared to untreated HCEC-12V4 (12V4) and asterisks (*) represent statistically significant differences between the untreated cell populations. Data are presented as the means ± SEM of *n* = 36 from four independent experiments (**b**,**c**) or *n* = 16 from two independent experiments (**d**,**e**). * *p* < 0.01, ** *p* < 0.001 and *** *p* = < 0.0001, one-way ANOVA with Tukey’s multiple comparisons test.

**Figure 6 membranes-12-00281-f006:**
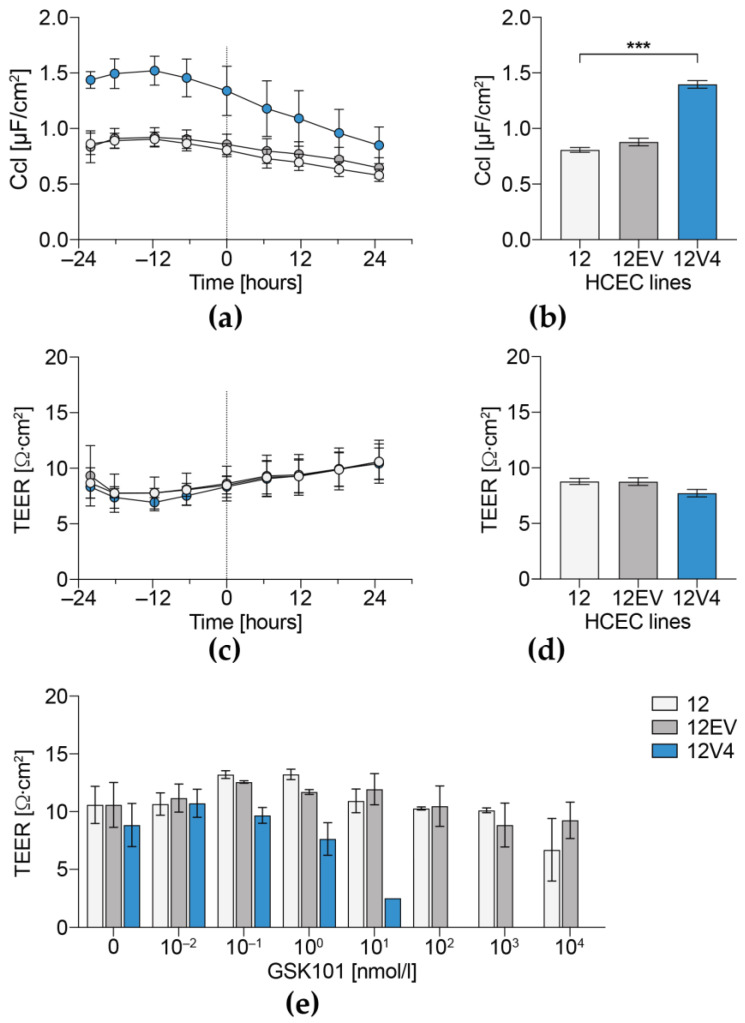
Dependence of GSK101-induced declines in transepithelial electrical resistance (TEER) and cellular capacitance (Ccl) on TRPV4 expression levels. Ccl (**a**,**b**) and TEER (**c**,**d**) were measured in unstimulated and stimulated cell populations for up to 48 h. Development of Ccl (**a**) and TEER (**c**) of the untreated control cells over time and comparison of stabilized TEER (**b**) and Ccl (**d**) values of all cells before stimulation and at time point 0 h (from **a**,**c**). The data are expressed as means ± SEM of *n* = 3 from 3 independent experiments. Development of TEER values after 24 h of stimulation with GSK101 (**e**). The data correspond to the means ± SEM of at least *n* = 6 from three independent experiments. The statistical analyses were carried out using one-way ANOVA followed by Tukey´s multiple comparisons test with *** *p* < 0.001.

**Figure 7 membranes-12-00281-f007:**
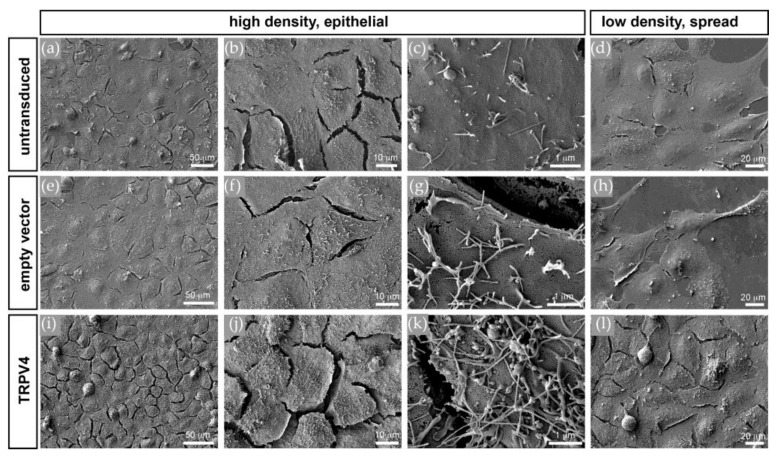
Scanning electron microscopy (SEM). Apical surfaces of confluent non-transduced (**a**–**d**), empty vector-transduced (**e**–**h**), and TRPV4-transduced (**i**–**l**) HCEC-12 cells at increasing magnifications from left to right. Note the differences in number and length of apical membrane protrusions, which is also apparent on the surfaces of less confluent or single cells (**d**,**h**,**l**).

**Table 1 membranes-12-00281-t001:** Primary and secondary antibodies.

Antigen ^1^	Host ^1^	Clone ^2^	Isotype	Supplier	Dilution, Method ^3^
TRPV4	rb	pc	IgG	Alomone Labs	WB 1:1000 ICC/IF 1:100
HA	rb	pc	IgG	Bethyl Laboratories	WB 1:1000 ICC/IF 1:100
HA	mu	HA.C5	IgG3	Abcam	WB 1:1000
ZO-1	mu	ZO1-1A12	IgG1	Invitrogen	ICC 1:100
y-tubulin	mu	GTU-88	IgG	Sigma	WB 1:5000
Na^+^/K^+^-ATPase	mu	464.6	IgG	Abcam	ICC 1:200
Isotype control	rb		IgG	Invitrogen	ICC/IF 1:100
Isotype control	mu		IgG	Invitrogen	ICC/IF 1:500
HRP-anti mu	gt	pc	IgG	Dako	WB 1:1000
HRP-anti rb	sw	pc	IgG	Dako	WB 1:1000
Alexa Fluor 488	gt	pc	IgG	Invitrogen	ICC/IF 1:800
Anti rb-IgG	gt	pc	IgG	Invitrogen	ICC 1:800
Alexa Fluor 568	rb	pc	IgG	Alomone Labs	WB 1:1000 ICC/IF 1:100
Anti mu-IgG	rb	pc	IgG	Bethyl Laboratories	WB 1:1000 ICC/IF 1:100

^1^ mu: mouse, rb: rabbit, gt: goat, sw: swine; ^2^ pc: polyclonal; ^3^ WB: Western blotting, ICC/IF: immunocytochemistry/immunofluorescence.

## Data Availability

The data presented in this study are available on request from the corresponding authors.

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
