# Peer review of "TRPV4 Stimulation Level Regulates Ca2+-Dependent Control of Human Corneal Endothelial Cell Viability and Survival"

_membranes, 2022, doi:10.3390/membranes12030281_

Round 1
Reviewer 1 Report
Comments for the authors:
In this paper, the authors investigated the TRPV4 stimulation effect on the Ca2+ -dependent control of human corneal endothelial cell vitality and survival. This study is well-designed, interesting, and contains novelty.
Reviewer 2 Report
The role of TRPV4 in human corneal endothelium cells is not fully understood. The authors explored the TRPV4 gene and protein overexpression on the viability and survival of HCEC. They demonstrate that TRPV4 the gene and protein overexpression in lentivirally transduced 12V4. Overexpression of TRPV4 did not alter the trans-endothelial electrical resistance (TEER) but increased their capacitance. Scanning electron microscopy showed villus-like protrusions only in the 12V4 cells. When they increased channel activity with GSK101, which induced calcium transients, cell death increased in 12V4 cells, it also reduced the TEER. On the opposite, the channel blocker RN1734 did not alter cell morphology, metabolic activity, or vitality.
The article is clear and interesting.
I could not find the patch-clamp experiments that they described in the statistical analysis. The observation of the currents is the most powerful tool to understand the behavior of the TRPV4 channel. I want to see the registers. It would be very interesting to see the channel behavior in a hypo-osmolar solution.
It would be important to contrast the results obtained in a cell line to the real corneal endothelium in the discussion and to scheme a model of the behavior of the channel.
A minor, reduce the number after the point, you only need two or three (lines 396-400).
Also, the term vitality is not entirely clear in the manuscript.
Reviewer 3 Report
General comments:
This manuscript by Donau et al. aims at clarifying the functional role of TRPV4 in corneal epithelium. The authors generate TRPV4 overexpressing human corneal epithelial cell populations (HCE) by lentiviral-mediated gene transfer. These populations expressed a HA-tagged TRPV4 protein at rather high levels. The phenotype of these cells was carefully characterized, and the authors demonstrate features corresponding to overexpression of a Ca2+ permeable channel. The HA-tagged protein was targeted significantly to the plasma membrane and stimulation with a synthetic agonist evoked large Ca2+ entry signals associated with downstream events expected for excessive Ca2+ loading such as metabolic changes and reduced survival. In addition, the authors present evidence for a distinct change in epithelial morphology that is potentially related to TRPV4 overexpression. Overall the discussion is lengthy and appears rather speculative. There are many open questions remaining, as outlined below. I think, to this end, the overexpression approach bears a high potential of miss- or overinterpretation. I suggest to revise the study to answer essential open questions and focus a bit more on the aspect of TRPV4-mediated control of cell morphology.
Main points:
- The authors generate a quite high level of TRPV4 overexpression in HCEC. This scenario might not correspond to any pathophysiological situation. What is the evidence for such levels of TRPV4 overexpression in HCE being disease relevant? Any reports on elevated TRPV4 expression in a diseased state of corneal epithelium?
- Is the overexpression model relevant for pathophysiology? Please discuss in more detail with critical consideration of the following: The majority of the generated channel complexes may be rather unique /artificial especially in terms of complex stoichiometry and subcellular targeting. The authors indeed analyze plasma membrane (PM) expression (Figure 3) but this analysis is rather incomplete. There is some evidence for PM targeting but this should be demonstrated more convincingly by higher resolution microscopy, use of PM markers, line scans to demonstrate the fluorescence distribution over cell compartments. How does this compare to the endogenous channels (immunocytochemistry)?
- Along the same line: It is somehow inconsistent and puzzling that the overexpressed channel apparently lacks the expected interaction with ATP1B1, K+/Na+ ATPase (Figure 3). Do overexpressed proteins behave differently than the endogenous channels and why?
- Moreover, do the authors have any evidence that the overexpressed channel display full mechanosensitivity/osmosensing function? The channel function was tested only in terms of its sensitivity to a synthetic agonist!
- The authors present certainly convincing evidence for the principle Ca2+ channel function of the overexpressed TRPV4. Nonetheless, it would be good to see an electrophysiological characterization of the overexpression generated vs the endogenous TRPV4 conductance. I see some indication that electrophysiological experiments might have been planed of have even been performed (statistics). I would strongly suggest to include these data.
- I find the observation of a rather distinct morphological changes by TRPV4 overexpression as an indeed intriguing result. Is this phenotype (enhanced formation of villus like protrusions) indeed correlated with TRPV4 channel function and TRPV4-mediated Ca2+ entry? Can morphological change be prevented or suppressed by a longer period of channel block?
Minor points:
- Methods section (statistcs) includes the statement “For patch-clamp and fluorescence data analyses…” I do not see electrophysiological data in the results.
- Please state more clearly how many stably transduced cell populations were generated and used in each experimental setting.
- The discussion is too lengthy and speculative. Please shorten und tune down speculative parts.
Round 2
Reviewer 3 Report
The authors provide a carfully revised version of the manuscript. They did a very good job in amending the manuscript in response to the raised criticism.